Prognostic significance of C-reactive protein (CRP) and albumin-based biomarker in patients with breast cancer receiving chemotherapy

Hutajulu Susanna Hilda 1
Astari Yufi Kartika 2
Ucche Meita 3
Kertia Nyoman nyoman.kertia@ugm.ac.id 4
Subronto Yanri Wijayanti 5
Paramita Dewi Kartikawati 6
Choridah Lina 7
Ekaputra Ericko 8
Widodo Irianiwati 9
Suwardjo Suwardjo 10
Hardianti Mardiah Suci 1
Taroeno-Hariadi Kartika Widayati 1
Purwanto Ibnu 1
Kurnianda Johan 1
1 Division of Hematology and Medical Oncology, Department of Internal Medicine, Faculty of Medicine, Public Health and Nursing, Universitas Gadjah Mada/Dr. Sardjito General Hospital , Sleman , Yogyakarta Special Region , Indonesia
2 Division of Hematology and Medical Oncology, Department of Internal Medicine, Dr. Sardjito General Hospital , Sleman , Yogyakarta Special Region , Indonesia
3 Study Program of Subspecialty, Division of Hematology and Medical Oncology, Department of Internal Medicine, Faculty of Medicine, Public Health and Nursing, Universitas Gadjah Mada/Dr. Sardjito General Hospital , Sleman , Yogyakarta Special Region , Indonesia
4 Division of Rheumatology, Department of Internal Medicine, Faculty of Medicine, Public Health and Nursing, Universitas Gadjah Mada/Dr. Sardjito General Hospital , Sleman , Yogyakarta Special Region , Indonesia
5 Division of Tropical Medicine and Infectious Diseases, Department of Internal Medicine, Faculty of Medicine, Public Health and Nursing, Universitas Gadjah Mada/Dr. Sardjito General Hospital , Sleman , Yogyakarta Special Region , Indonesia
6 Department of Histology and Cell Biology, Faculty of Medicine, Public Health and Nursing, Universitas Gadjah Mada , Sleman , Yogyakarta Special Region , Indonesia
7 Division of Radiodiagnosis, Department of Radiology, Faculty of Medicine, Public Health and Nursing, Universitas Gadjah Mada/Dr. Sardjito General Hospital , Sleman , Yogyakarta Special Region , Indonesia
8 Division of Radiotherapy, Department of Radiology, Faculty of Medicine, Public Health and Nursing, Universitas Gadjah Mada/Dr. Sardjito General Hospital , Sleman , Yogyakarta Special Region , Indonesia
9 Department of Anatomical Pathology, Faculty of Medicine, Public Health and Nursing, Universitas Gadjah Mada , Sleman , Yogyakarta Special Region , Indonesia
10 Division of Surgical Oncology, Department of Surgery, Faculty of Medicine, Public Health and Nursing, Universitas Gadjah Mada/Dr. Sardjito General Hospital , Sleman , Yogyakarta Special Region , Indonesia
Samuel Stephen
Electronic publication date: 2025 May 21
Publication date: 2025
Volume: 13
Electronic Location ID: e19319
Received 2024 Dec 11; Accepted 2025 Mar 24
Copyright: ©2025 Hutajulu et al.
Copyright year: 2025
Copyright holder: Hutajulu et al.
License: This is an open access article distributed under the terms of the Creative Commons Attribution License, which permits unrestricted use, distribution, reproduction and adaptation in any medium and for any purpose provided that it is properly attributed. For attribution, the original author(s), title, publication source (PeerJ) and either DOI or URL of the article must be cited.
License URL: https://creativecommons.org/licenses/by/4.0/

Keywords: Breast neoplasms, C-reactive protein, Albumin, Survival, Mortality

Funding: Academic Excellence grant from Universitas Gadjah Mada 4344/UN1/DITLIT/PT.01.03/2024 This work was supported by Academic Excellence grant from Universitas Gadjah Mada (grant number: 4344/UN1/DITLIT/PT.01.03/2024). The funders had no role in study design, data collection and analysis, decision to publish, or preparation of the manuscript.

==============================
Background

Breast cancer patients with similar clinicopathologic characteristics may experience varied outcomes. This urges an increased effort to investigate other prognostic factors. C-reactive protein (CRP)-to-albumin ratio (CAR) is an inflammatory and nutritional biomarker that has been well studied and reported to have an impact on the survival of patients with diverse types of cancer, but limitedly in breast cancer. Therefore, this study aimed to investigate the prognostic significance of CAR in local patients with breast cancer.

Methods

This study included 202 stage I–IV breast cancer patients receiving first-line chemotherapy. We calculated inflammatory and nutritional biomarkers including CAR, neutrophil-to-lymphocyte ratio (NLR), platelet-to-lymphocyte ratio (PLR), lymphocyte-to-monocyte ratio (LMR), systemic immune-inflammation index (SII), systemic inflammation response index (SIRI), pan-immune-inflammation value (PIV), and prognostic nutrition index (PNI) before treatment. The Kaplan-Meier with log-rank test and Cox proportional hazard regression were used to analyze the prognostic role of clinicopathologic factors and biomarkers on disease-free survival (DFS), progression-free survival (PFS), and overall survival (OS).

Results

The median follow-up period was 46 months (1–77 months). The 3-year DFS and 3-year OS in patients with high CAR (CAR > 1.5) were significantly lower than those with low CAR (CAR ≤ 1.5) (47.0% vs 68.9%, P = 0.022 and 59.5% vs 78.6%, P = 0.009, respectively). Multivariate analysis showed high CAR as prognostic factors for poor DFS (HR 2.10, 95% confidence interval/CI [1.10–3.99], P = 0.023) and OS (HR 2.16, 95% CI [1.27–3.68], P = 0.005), but not for PFS (HR 1.43, 95% CI [0.73–2.80], P = 0.293). In addition, more advanced stage and HER2 positive were correlated with unfavorable DFS and OS, older age predicted poor DFS, and stage was the only prognostic factor of PFS (all P values < 0.05).

Conclusion

Besides age, stage, and molecular subtypes that have been widely observed to have impact on the survival of breast cancer patients, CAR was demonstrated as a promising prognostic marker in our local patients. A high CAR at diagnosis was associated with unfavorable DFS and OS, which can aid in identifying patients at risk and guide personalized treatment planning.

Introduction

Breast cancer is the most prevalent malignancy with 66,271 new cases in Indonesia based on GLOBOCAN 2022 data. This cancer accounted for 30.1% of malignancy among women and its mortality ranked third (9.3%) after lung and liver cancer (Ferlay et al., 2024) in both sexes. The 5-year survival was only 51% across all stages and 71% among stage III disease (Sinaga et al., 2018; Deanasa, Umar & Fitri, 2022). Breast cancer survival is affected by classic prognostic factors including tumor size, tumor grade, axillary lymph node involvement, disease stage, estrogen receptor (ER) status, and human epidermal growth factor receptor 2 (HER2) overexpression (Galea et al., 1992; Goldhirsch et al., 2005). However, patients with similar clinicopathologic characteristics may experience varied clinical outcomes. Thus, there is a demand to increase exploration of other prognostic factors in order to enable a more individualized treatment, and eventually improve patient outcomes (Susini, Biglia & Bounous, 2022).

It has been observed that the response of the body to cancer greatly corresponds with inflammation and wound healing, during which inflammatory cells and cytokines in tumors play essential roles in tumor development, progression and immunosuppression (Balkwill & Mantovani, 2001). Many inflammatory and nutritional biomarkers were demonstrated as independent prognostic factors in breast cancer, including neutrophil-to-lymphocyte ratio (NLR), platelet-to-lymphocyte ratio (PLR), lymphocyte-to-monocyte ratio (LMR), C-reactive protein (CRP), systemic immune-inflammation index (SII), systemic inflammation response index (SIRI), pan-immune-inflammation value (PIV), prognostic nutrition index (PNI), and CRP-to-albumin ratio (CAR) (Zhou et al., 2019; Hua et al., 2020; Savioli et al., 2022; Cheng et al., 2024; Zhang & Cheng, 2024). Among these biomarkers, CAR had the greatest discriminating potential in predicting survival outcomes in various types of malignancies such as colorectal, esophageal, and pancreatic cancer (Kim et al., 2020; Matsunaga et al., 2020; Suzuki et al., 2020). CAR reflects both the inflammatory and nutritional status. C-reactive protein (CRP) is an acute-phase protein produced in the liver, which can promote tumor cell invasion, angiogenesis, and metastasis in the inflammatory milieu. Elevated pre-treatment CRP levels have been observed to correlate with tumor aggressiveness, poor treatment response, and unfavorable outcomes in breast cancer patients (Allin et al., 2011; Asegaonkar et al., 2015). Elevated CRP levels are also linked to lower albumin levels because albumin synthesis rate in the liver is suppressed. Serum albumin is a frequently used nutritional status biomarker where nutritional deficiency impairs immune system, reduces the effectiveness of treatment, and increases the risk of unfavorable outcomes in cancer patients (Zhou et al., 2019).

Although there is growing evidence of the prognostic role of CAR, particularly in gastrointestinal malignancies (Xu et al., 2017; Kim et al., 2020; Matsunaga et al., 2020; Suzuki et al., 2020), studies in breast cancer are limited (Zhou et al., 2019; Liu, Guo & Zhang, 2021). Thus, this study aimed to investigate the relationship of CAR serum level as an inflammatory and nutritional biomarker with the survival of local patients with stage I–IV breast cancer.

Materials & Methods

Study design and participants

This retrospective cohort study was authorised by the Medical and Health Research and Ethics Committee, Faculty of Medicine, Public Health and Nursing, Universitas Gadjah Mada (reference number: KE/FK/0417/EC/2018). Each participant provided written and signed informed consent before study enrolment. From July 2018 to March 2022, women with breast cancer undergoing first-line chemotherapy, either on adjuvant, neoadjuvant, or palliative setting, in the Hematology and Medical Oncology Division, “Tulip”/Integrated Cancer Clinic, Dr Sardjito General Hospital Yogyakarta, were included in this study. Patients who were enrolled in the study had been pathologically diagnosed with stage I–IV breast cancer, aged not less than 18 years, had an Eastern Cooperative Oncology Group (ECOG) performance status of ≤2, without any history of previous chemotherapy, and terminal illness. Patient was excluded if she failed to receive chemotherapy and had not reached the three-year follow-up duration.

Patients with stage I–III breast cancer were assigned to locoregional treatment followed by adjuvant chemotherapy, except for patients considered to receive pre-operative or neoadjuvant chemotherapy. Candidates for neoadjuvant chemotherapy included inflammatory breast cancer, cT4 tumor, cN3 nodal disease, or HER2-positive and triple-negative breast cancer with ≥cT2 or ≥cN1. Patients with stage IV breast cancer were assigned to receive locoregional treatment and/or palliative chemotherapy (Gradishar et al., 2024). The chemotherapy regimen for adjuvant and neoadjuvant setting was anthracycline (doxorubicin or epirubicin) and cyclophosphamide followed by taxane (docetaxel or paclitaxel) every 3 weeks. The chemotherapy regimen for palliative setting was taxane, as monotherapy or in combination with carboplatin, or capecitabine. The endpoints of this study were disease-free survival (DFS) for patients receiving adjuvant chemotherapy, progression-free survival (PFS) for patients receiving neoadjuvant or palliative chemotherapy, and overall survival (OS) for all included participants.

Of the 250 eligible patients, 48 were excluded because they did not receive chemotherapy (n = 36) or the follow-up duration had not reached 3 years (n = 12). A total of 202 patients with stage I–IV breast cancer were included in the study and later in the OS analysis. One hundred and twenty-nine patients received adjuvant chemotherapy and were included in the DFS analysis. Seventy-three patients received neoadjuvant and palliative chemotherapy and were included in the PFS analysis (Fig. 1).

Figure 1 Flowchart of subject’s inclusion to the study and final analysis.

Data collection and definition

Patient clinical and pathological data collection included age at diagnosis (<50 years vs ≥50 years), the tumor, nodes, metastasis (TNM) stage according to the criteria of the American Joint Committee on Cancer (AJCC) 8th edition, the estrogen receptors (ER), progesterone receptors (PR), and HER2 status (negative vs positive), and the chemotherapy intention (adjuvant, neoadjuvant, or palliative). The complete blood cell count, including neutrophil (/µl), lymphocyte (/µl), monocyte (/µl), and platelet (/µl) counts, as well as serum albumin (g/dl) were routinely measured for each patient. In addition, 21 ml peripheral blood samples were obtained before the first chemotherapy cycle. After centrifugation, plasma samples were aliquoted and stored at −80 °C in Biobank, Faculty of Medicine, Public Health, and Nursing, Universitas Gadjah Mada. CRP serum levels were analyzed using a quantitative Enzyme-linked Immunosorbent Assay (ELISA) (Calbiotech, El Cajon, California, Catalog No. CR120C) in the Integrated Research laboratory, Faculty of Medicine, Public Health, and Nursing, Universitas Gadjah Mada by trained laboratory staff. Thus, with the collected laboratory data, we analyzed inflammation and nutritional biomarkers including NLR, PLR, LMR, SII, SIRI, PIV, PNI, and CAR. Each biomarker’s definition and formula were described in Table 1.

Cut-off values for inflammatory and nutritional biomarkers

We performed receiver operating curve (ROC) analysis to determine optimal cut-off levels of all biomarkers using the Liu method (Liu, 2012). DFS, PFS, and OS rates are the most appropriate measures of prognosis. Considering it is unusual to designate more than one cut-off level in the same biomarker, we chose a 3-year OS as a stratifying point. Determined optimal cut-offs of NLR, PLR, LMR, SII, SIRI, PIV, PNI, and CAR were 1.995, 145.5, 0.265, 774.8, 1.11, 331.8, 54.06, and 1.5, respectively (Table 2). Patients were further grouped based on these cut-off values.

To evaluate the biomarker’s predictive ability, we measured and compared the area under the ROC curve (AUC) of each biomarker. Among all inflammatory and nutritional biomarkers, CAR showed the highest AUC of 0.608 (Table 2). Thus, patients’ baseline characteristics were further compared between CAR groups. Patients with CAR ≤1.5 were grouped as low CAR and patients with CAR >1.5 were grouped as high CAR.

Follow-up

All study participants underwent three-monthly follow-up after chemotherapy for the first year, and every six months afterwards. Follow-up included clinical examination, laboratory tests, breast ultrasonography, abdominal ultrasonography, chest X-ray, bone survey or bone scan, and other examinations as deemed appropriate. The final follow-up ended on August 2024, through medical records and telephone follow-up. DFS and PFS were referred to the time from pathological diagnosis to the first detection of local recurrence, disease progression, or death, or to the last follow-up. OS was defined as the time from pathological diagnosis to death from any cause or to the last follow-up.

Table 1 Eight biomarkers evaluated in this study.

Biomarker’s name	Biomarker’s formula	
Neutrophil-to-lymphocyte ratio (NLR)	Neutrophil/lymphocyte	
Platelet-to-lymphocyte ratio (PLR)	Platelet/lymphocyte	
Lymphocyte-to-monocyte ratio (LMR)	Lymphocyte/monocyte	
Systemic immune-inflammation index (SII)	Neutrophil × platelet/lymphocyte	
Systemic inflammation response index (SIRI)	Neutrophil × monocyte/lymphocyte	
Pan-immune-inflammation value (PIV)	Neutrophil × platelet × monocyte/lymphocyte	
Prognostic nutrition index (PNI)	(10 × Albumin) + (0.005 × lymphocyte)	
CRP-to-albumin ratio (CAR)	CRP/albumin	
Notes.

CRP C-reactive protein

Table 2 Comparison of area under curve (AUC) of evaluated biomarkers for overall survival.

Biomarkers	Cut-off	AUC	95% CI	
NLR	1.995	0.513	0.43–0.59	
PLR	145.5	0.473	0.39–0.56	
LMR	0.265	0.576	0.49–0.66	
SII	774.8	0.465	0.38–0.55	
SIRI	1.11	0.475	0.39–0.56	
PIV	331.8	0.490	0.40–0.57	
PNI	54.06	0.561	0.48–0.64	
CAR	1.5	0.608	0.52–0.69	
Notes.

CI confidence interval

NLR neutrophil-to-lymphocyte ratio

PLR platelet-to-lymphocyte ratio

LMR lymphocyte-to-monocyte ratio

SII systemic immune-inflammation index

SIRI systemic inflammation response index

PIV pan-immune inflammation value

PNI prognostic nutritional index

CAR C-reactive protein-to-albumin ratio

Statistical analysis

The categorical variables were presented as frequencies with percentages and the numerical variables were presented as median with interquartile range (IQR). The survival status was estimated using the Kaplan–Meier curves with log-rank test. Clinicopathological factors and biomarkers associated with survival outcomes were analyzed using univariate and multivariate Cox proportional hazard regression. Clinicopathological factors and biomarkers with P < 0.05 in the univariate analysis and variable of interest in this current study, CAR, were entered into multivariate analysis for DFS, PFS, and OS. A P < 0.05 was considered statistically significant. The statistical analyses were performed with STATA version 17 (Stata Corp., College Station, TX).

Results

Baseline characteristics

The median age was 51 years (IQR 45.4–57.6 years). Most of patients were diagnosed with stage III breast cancer (42.1%), with ER positive (61.9%), and HER2 negative (67.3%). Compared between the low CAR and high CAR groups, patients’ baseline characteristics showed significant differences in terms of stage and chemotherapy intention. High CAR was more frequent in patients with stage IV (35.5% vs 12.5%, P = 0.002), those receiving neoadjuvant treatment, and those receiving palliative chemotherapy (10.5% vs 6.8% and 35.5% vs 12.5%, P = 0.001) (Table 3).

Table 3 Baseline characteristics of the subjects included in the study by CAR status.

Characteristics	All subjects
(n = 202)	CAR ≤1.5
(n = 88)	CAR > 1.5
(n = 76)	P	
Age (Median (IQR))	51.3 (45.4–57.6)				
<50 years	73 (44.5)	45 (51.1)	28 (36.8)	0.066	
≥50 years	91 (55.5)	43 (48.9)	48 (63.2)		
Stage					
I–II	60 (29.7)	30 (34.1)	20 (26.3)	0.002	
III	85 (42.1)	47 (53.4)	29 (38.2)		
IV	57 (28.2)	11 (12.5)	27 (35.5)		
ER					
Negative	74 (36.6)	35 (39.8)	27 (35.5)	0.576	
Positive	125 (61.9)	53 (60.2)	49 (64.5)		
Unknown	3 (1.5)				
PR					
Negative	93 (46.0)	45 (51.1)	33 (43.4)	0.324	
Positive	106 (52.5)	43 (48.9)	43 (56.6)		
Unknown	3 (1.5)				
HER2					
Negative	136 (67.3)	57 (66.3)	52 (69.3)	0.679	
Positive	60 (29.7)	29 (33.7)	23 (30.7)		
Unknown	6 (3.0)				
Biomarkers (Median (IQR))					
NLR	2.48 (1.78–3.47)	2.41 (1.70–3.46)	2.47 (1.84–3.35)	0.984	
PLR	157 (117–214)	161 (124–208)	143 (112–207)	0.164	
LMR	0.26 (0.20–0.36)	0.26 (0.19–0.34)	0.26 (0.19–0.35)	0.796	
SII	720 (494–1075)	681 (472–1025)	753 (489–1087)	0.582	
SIRI	1.20 (0.78–1.73)	1.06 (0.71–1.52)	1.23 (0.82–1.74)	0.202	
PIV	325 (211–569)	294 (202–538)	384 (224–583)	0.218	
PNI (n = 191)	52.8 (47.9–56.8)	53.9 (50.3–56.9)	53.3 (47.9–57.4)	0.388	
CAR (n = 164)	1.23 (0.39–2.67)	–	–	–	
Chemotherapy intention					
Adjuvant	129 (63.9)	71 (80.7)	41 (54.0)	0.001	
Neoadjuvant	16 (7.9)	6 (6.8)	8 (10.5)		
Palliative	57 (28.2)	11 (12.5)	27 (35.5)		
Notes.

IQR interquartile range

ER estrogen receptors

PR progesterone receptors

HER2 human epidermal growth factor receptor 2

NLR neutrophil-to-lymphocyte ratio

PLR platelet-to-lymphocyte ratio

LMR lymphocyte-to-monocyte ratio

SII systemic immune-inflammation index

SIRI systemic inflammation response index

PIV pan-immune inflammation value

PNI prognostic nutritional index

CAR C-reactive protein-to-albumin ratio

Patients’ survival by clinicopathologic factors, inflammatory, and nutritional investigated biomarkers

Overall, the median follow-up of patients in this study was 46 months (1–77 months). The median DFS and PFS were 73.2 and 13.1 months, while the median OS was not reached. The 3-year DFS, PFS, and OS are 62.9%, 16.9%, and 65.7% (Figs. 2A–2C). Kaplan–Meier survival curves and log-rank test revealed that cases with advanced stage (stage III or IV) (P = 0.004 and P < 0.001), HER2 positive (P < 0.001 and P = 0.023), and high CAR (P = 0.022 and P = 0.009) had a significantly poorer 3-year DFS (Figs. 3B–3D) and OS (Figs. 3J–3L) compared with their counterparts. In addition, age ≥50 years (P < 0.001) was significantly associated with poorer DFS (Fig. 3A). The 3-year DFS and 3-year OS in patients with high CAR were significantly lower than those with low CAR (47.0% vs 68.9%, P = 0.022 and 59.5% vs 78.6%, P = 0.009) (Figs. 3D and 3L). Stage was the only parameter related to 3-year PFS in which cases with stage IV disease (P = 0.007) had unfavorable survival (Fig. 3F). The 3-year PFS of patients with high and low CAR was similar (20.6% vs 23.5%, P = 0.333) (Fig. 3H). No significant differences in 3-year DFS, PFS, and OS were observed in cases having different levels of other inflammatory and nutritional biomarkers (Figs. S1–S3).

Figure 2 Breast cancer survival.

(A) Kaplan–Meier curves for disease-free survival (DFS) in 129 breast cancer patients receiving adjuvant chemotherapy. (B) Kaplan–Meier curves for progression-free survival (PFS) in 73 breast cancer patients receiving neoadjuvant or palliative chemotherapy. (C) Kaplan–Meier curves for overall survival (OS) in 202 breast cancer patients receiving chemotherapy.

Figure 3 Relationship between clinicopathological factors and CAR and breast cancer survival.

(A–D) Kaplan–Meier curves for 3-year disease-free survival (DFS) based on age, stage, human epidermal receptor 2 (HER2) status, and C-reactive protein to albumin ratio (CAR). (E–H) Kaplan Meier curves for 3-year progression-free survival (PFS) based on age, stage, HER2 status, and CAR. (I–L) Kaplan Meier curves for 3-year overall survival (OS) based on age, stage, HER2 status, and CAR.

Prognostic significance of clinicopathologic factors, inflammatory, and nutritional biomarkers

Analyses of DFS prognostic factors were summarized in Table 4. According to multivariate analysis, high CAR was a prognostic factor for unfavorable DFS (HR 2.10, 95% CI [1.10–3.99], P = 0.023). Age (HR 2.32, 95% CI [1.21–4.48], P = 0.012), stage (HR 3.07, 95% CI [1.58–5.95], P = 0.001), and HER2 status (HR 2.99, 95% CI [1.60–5.61], P = 0.001) were also predictors for DFS.

Table 4 Univariate and multivariate analysis predicting disease-free survival (DFS).

Factor	n	Univariate (n = 129)	Multivariate ((n = 110)	
		HR	95% CI	P	HR	95% CI	P	
Age								
<50 years	57	Ref			Ref			
≥50 years	72	2.78	1.53–5.04	0.001	2.32	1.21–4.48	0.012	
Stage								
I–II	57	Ref			Ref			
III	72	2.41	1.34–4.31	0.003	3.07	1.58–5.95	0.001	
ER								
Negative	50	Ref			Ref			
Positive	79	0.58	0.34–0.99	0.047	0.69	0.37–1.30	0.254	
PR								
Negative	61	Ref						
Positive	68	0.65	0.38–1.11	0.118				
HER2 (n = 127)								
Negative	91	Ref			Ref			
Positive	36	2.54	1.48–4.37	0.001	2.99	1.60–5.61	0.001	
NLR								
≤1.995	53	Ref						
>1.995	76	0.87	0.51–1.48	0.607				
PLR								
≤145.5	62	Ref						
>145.5	67	1.07	0.63–1.82	0.803				
LMR								
≥0.265	54	Ref						
<0.265	75	1.45	0.84–2.52	0.185				
SII								
≤774.8	80	Ref						
>774.8	49	0.97	0.56–1.67	0.913				
SIRI								
≤1.11	66	Ref						
>1.11	63	0.67	0.39–1.15	0.147				
PIV								
≤331.8	71	Ref						
>331.8	58	0.78	0.45–1.33	0.355				
PNI (n = 120)								
≥54.06	65	Ref						
<54.06	55	0.67	0.38–1.17	0.161				
CAR (n = 112)								
≤1.5	71	Ref			Ref			
>1.5	41	1.61	0.92–2.83	0.097	2.10	1.10–3.99	0.023	
Notes.

HR Hazard Ratio

CI confidence interval

ER estrogen receptors

PR progesterone receptors

HER2 human epidermal growth factor receptor 2

NLR neutrophil-to-lymphocyte ratio

PLR platelet-to-lymphocyte ratio

LMR lymphocyte-to-monocyte ratio

SII systemic immune-inflammation index

SIRI systemic inflammation response index

PIV pan-immune inflammation value

PNI prognostic nutritional index

CAR C-reactive protein-to-albumin ratio

Analyses of PFS prognostic factors were summarized in Table 5. In univariate analysis of PFS, stage was the only significant prognostic factor. The multivariate analysis also showed that stage IV was associated with poor PFS (HR 2.45, 95% CI [1.12–5.37], P = 0.025), while CAR was not (HR 1.43, 95% CI [0.73–2.80], P = 0.293).

Table 5 Univariate and multivariate analysis predicting progress-free survival (PFS).

Factor	n	Univariate (n = 73)	Multivariate (n = 52)	
		HR	95% CI	P	HR	95% CI	P	
Age								
<50 years	35	Ref						
≥50 years	38	0.83	0.50–1.38	0.467				
Stage								
II–III	16	Ref			Ref			
IV	57	2.24	1.13–4.47	0.021	2.45	1.12–5.37	0.025	
ER (n = 70)								
Negative	24	Ref						
Positive	46	0.97	0.55–1.70	0.922				
PR (n = 70)								
Negative	32	Ref						
Positive	38	1.03	0.60–1.74	0.920				
HER2 (n = 69)								
Negative	45	Ref						
Positive	24	1.04	0.59–1.82	0.883				
NLR								
≤1.995	16	Ref						
>1.995	57	1.14	0.61–2.10	0.685				
PLR								
≤145.5	25	Ref						
>145.5	48	1.08	0.63–1.84	0.773				
LMR								
≥0.265	45	Ref						
<0.265	28	0.72	0.42–1.23	0.227				
SII								
≤774.8	31	Ref						
>774.8	42	1.26	0.75–2.11	0.385				
SIRI								
≤1.11	24	Ref						
>1.11	49	1.24	0.71–2.14	0.447				
PIV								
≤331.8	32	Ref						
>331.8	41	1.63	0.97–2.75	0.066				
PNI (n = 70)								
≥54.06	16	Ref						
<54.06	54	1.25	0.67–2.34	0.475				
CAR (n = 52)								
≤1.5	17	Ref			Ref			
>1.5	35	1.48	0.76–2.88	0.249	1.43	0.73–2.80	0.293	
Notes.

HR Hazard Ratio

CI confidence interval

ER estrogen receptors

PR progesterone receptors

HER2 human epidermal growth factor receptor 2

NLR neutrophil-to-lymphocyte ratio

PLR platelet-to-lymphocyte ratio

LMR lymphocyte-to-monocyte ratio

SII systemic immune-inflammation index

SIRI systemic inflammation response index

PIV pan-immune inflammation value

PNI prognostic nutritional index

CAR C-reactive protein-to-albumin ratio

Analyses of OS prognostic factors were summarized in Table 6. In the univariate and multivariate analysis, high CAR was a prognostic factor of unfavorable OS (HR 2.11, 95% CI [1.24–3.57], P = 0.005 and HR 2.16, 95% CI [1.27–3.68], P = 0.005). Stage (HR 9.42, 95% CI [2.94–30.25], P < 0.001) and HER2 status (HR 2.05, 95% CI [1.17–3.56], P = 0.011) were also independent predictors for OS.

Table 6 Univariate and multivariate analysis predicting overall survival (OS).

Factor	n	Univariate (n = 202)	Multivariate (n = 161)	
		HR	95% CI	P	HR	95% CI	P	
Age								
<50 years	92	Ref						
≥50 years	110	1.16	0.74–1.81	0.520				
Stage								
I–II	60	Ref			Ref			
III–IV	142	11.32	4.14–30.96	<0.001	9.42	2.94–30.25	<0.001	
ER (n = 199)								
Negative	74	Ref						
Positive	125	0.70	0.45–1.10	0.126				
PR (n = 199)								
Negative	93	Ref						
Positive	106	0.72	0.46–1.12	0.149				
HER2 (n = 196)								
Negative	136	Ref			Ref			
Positive	60	1.88	1.19–2.95	0.006	2.05	1.17–3.56	0.011	
NLR								
≤1.995	69	Ref						
>1.995	133	1.45	0.89–2.34	0.130				
PLR								
≤145.5	87	Ref						
>145.5	115	1.24	0.79–1.93	0.351				
LMR								
≥0.265	99	Ref						
<0.265	103	0.83	0.54–1.29	0.410				
SII								
≤774.8	111	Ref						
>774.8	91	1.27	0.82–1.97	0.287				
SIRI								
≤1.11	90	Ref						
>1.11	112	1.19	0.77–1.86	0.429				
PIV								
≤331.8	103	Ref						
>331.8	99	1.20	0.77–1.86	0.415				
PNI (n = 190)								
≥54.06	81	Ref			Ref			
<54.06	109	1.89	1.17–3.06	0.009	1.27	0.72–2.24	0.402	
CAR (n = 164)								
≤1.5	88	Ref			Ref			
>1.5	76	2.11	1.24–3.57	0.005	2.16	1.27–3.68	0.005	
Notes.

HR Hazard Ratio

CI confidence interval

ER estrogen receptors

PR progesterone receptors

HER2 human epidermal growth factor receptor 2

NLR neutrophil-to-lymphocyte ratio

PLR platelet-to-lymphocyte ratio

LMR lymphocyte-to-monocyte ratio

SII systemic immune-inflammation index

SIRI systemic inflammation response index

PIV pan-immune inflammation value

PNI prognostic nutritional index

CAR C-reactive protein-to-albumin ratio

Discussion

Globally, breast cancer is the most prevalent malignancy in females, with significant variations in survival rates across different regions. The 5-year survival of breast cancer was over 85% in the USA, Canada, European countries, Japan, and South Korea (Allemani et al., 2018). In Asia region, Thailand, India, and Malaysia had low 5-year survival (68.7%, 66.1%, 65%, respectively) (Soerjomataram et al., 2023), but still higher than Indonesia (51%) (Sinaga et al., 2018). The present study reported breast cancer 3-year survival of 65.7%, which is lower than Thailand (71.4%) and Malaysia (73.7%), as well as China (88.7%) and South Korea (93.5%) (Soerjomataram et al., 2023). Compared to high-income and even neighboring countries, low breast cancer survival in Indonesia calls for continuous efforts to improve breast cancer care in Indonesia.

There is growing evidence that breast cancer progression is significantly influenced by the inflammatory pathway (Kaur et al., 2019). In the inflammatory condition, the synthesis of CRP was increased but the production of albumin was suppressed. A low level of albumin may also reduce the tolerance of systemic anti-cancer treatment. Therefore, long-term elevated CRP and reduced albumin levels were predictors of poor survival (Nazha, 2015). CRP-to-albumin ratio (CAR), a biomarker related to inflammation and nutrition, has been demonstrated as a prognostic factor in various cancers including colorectal cancer (Ruan et al., 2023b), gastric cancer (Sakai et al., 2020), pancreatic cancer (Fujiwara et al., 2018), esophageal cancer (Ishibashi et al., 2018), and ovarian cancer (Komura et al., 2021). However, there are few studies on CAR in breast cancer, highlighting the clinical importance of this present study. This study revealed CAR as a prognostic factor in breast cancer patients receiving first-line chemotherapy. It was observed that a high CAR was correlated with poor DFS and OS, aligning with findings from previous studies (Zhou et al., 2019; Liu, Guo & Zhang, 2021; Ruan et al., 2023a).

The underlying causative association between CAR and breast cancer survival remains unclear but several hypotheses have been suggested. CRP, as the first component of CAR, is a well-known acute-phase protein in response to inflammation, trauma, or tissue damage. Even though breast cancer is rarely characterized by a significant histologic inflammation, CRP level might be moderately risen indicating a low-grade chronic inflammation (Asegaonkar et al., 2015). An inflammatory condition in the breast cancer tumor micro-environment results in pro-tumorigenic and pro-angiogenic response which is associated with disease progression, risk of recurrence, and poor outcome (Allin et al., 2011). The molecular pathway underlying an elevated CRP in inflammatory environments has been reported in breast epithelial cells (Kim et al., 2014). Sphingosine-1-phosphate (S1P) markedly activated CRP transcription, increasing the expression of CRP which is secreted into extracellular space. This S1P-induced CRP expression resulted in the upregulation of matrix metalloproteinase (MMP)-9 through extracellular signal-related kinase (ERKs), reactive oxygen species (ROS), and c-fos. Using a xenograft mice tumor model, S1P-induced CRP expression was demonstrated both in vitro and in vivo explaining their role in increasing invasive phenotype in breast cancer cells (Kim et al., 2014; Kim, Kim & Moon, 2023).

At the time of diagnosis, elevated CRP level reflects tumor aggressiveness, such as larger tumor size, more advanced stage, higher grade, or the presence of distant metastasis (Allin et al., 2011; Asegaonkar et al., 2015). An elevated CRP level will result in an elevated CAR value. In our study, we found that the proportion of patients with a high CAR was significantly higher in patients with stage IV compared to stage I–III disease. We expected that patients with high CAR at diagnosis are carrying elevated inflammatory levels and more severe breast cancer disease, resulting in more aggressive chemotherapy regimen consideration and requiring close attention to prevent poor survival.

The second component of CAR is albumin. In addition to evaluating patient’s nutritional status, serum albumin level might be an indicator that links to disease outcomes in many cancer types including breast cancer. The mechanism by which albumin level might impact cancer survival is expected due to its essential physiologic functions, such as maintaining plasma oncotic pressure, lowering pro-inflammatory fatty acids through its binding effect, and exerting antioxidant activity by scavenging ROS and limiting ROS production. These mechanisms suggest that albumin may inhibit tumor progression, thus improving survival (Tang, Li & Sun, 2024). A previous study on breast cancer cell line, MCF-7, demonstrated that albumin may affect cell proliferation by modulating the activation of autocrine growth regulatory factors (Laursen, Briand & Lykkesfeldt, 1990). However, despite several hypotheses mentioned above, the underlying biological link between elevated CAR and poor survival remains unexplored, warranting studies on chronic inflammation, hypoalbuminemia, and tumor aggressiveness.

Our observation on the significant role of CAR in DFS supported previous studies enrolling non-metastatic breast cancer in China (Zhou et al., 2019; Liu, Guo & Zhang, 2021). Zhou et al. (2019) included 200 resectable cases and reported a significantly reduced 5-year DFS among patients with high CAR compared to low CAR (59.8% vs 79.2%). Though these results are in line with our findings, the DFS rates are better than ours, in accordance with the aforementioned epidemiological difference. Liu, Guo & Zhang (2021) included 199 patients with stage I-III luminal B breast cancer and reported that high CAR was correlated with unfavorable DFS with a higher hazard when compared to that of Zhou et al. (2019) and ours (HR 4.35 vs HR 2.22 and HR 2.10). This might be due to the different numbers of variables included in the multivariate analysis, where Liu, Guo & Zhang (2021) was the highest. A more complex model with a limited sample size can lead to overfitting (Babyak, 2004). In terms of OS, our findings were also comparable to previous report recruiting 514 patients with breast cancer (HR 2.56) (Ruan et al., 2023a).

Our study demonstrating no significant association between high CAR and poor PFS supported previous observation in metastatic breast cancer (Kayikciouglu & Onder, 2022). However, there are conflicting findings on the prognostic role of CAR for PFS among other cancers. A study in 90 metastatic colorectal cancer cases showed no difference in 5-year relapse-free survival among such patients having high and low CAR (Ishii et al., 2022). On the contrary, Kim et al. (2020) found that high CAR was significantly associated with shorter PFS (HR 1.48, 95% CI [1.17–1.88], P < 0.001) in unresectable pancreatic cancer patients receiving palliative chemotherapy. Kim et al. (2020) used a markedly higher CAR cut-off compared to our study (3.85 vs 1.50). In addition, the median follow-up time from these studies was different, 46–48 months in breast cancer (Kayikciouglu & Onder, 2022), 44 months in colorectal cancer (Ishii et al., 2022), and 7 months in pancreatic cancer (Kim et al., 2020). The differences in the cut-off value for CAR, the selection of study subjects, and the follow-up time might explain these conflicting findings. Thus, further investigation is required to address this discrepancy.

Besides CAR, our study also showed that age, stage, and HER2 status were significant prognostic parameters in breast cancer. Advanced age and stage are the established and prominent prognostic factors in breast cancer survival, both at global and national levels (Mursyidah, Ashariati & Kusumastuti, 2019; Fernandes et al., 2023). This study reported that HER2 status was significantly associated with poor DFS and OS. HER2 positivity is well known to link to increased brain metastasis, disease recurrence, and mortality risk contributing to worse survival (Kang et al., 2023). Moreover, vast majority of our patients do not have access to anti-HER2 antibody such as trastuzumab because the drug is limitedly covered by the national insurance that supports almost all of our patients (Menteri Kesehatan Republik Indonesia, 2018).

The measurements of CRP and albumin are readily available, easy to measure, and comparably cheap. In addition, CAR has straightforward calculation and a simple cut-off. Thus, incorporating CAR assessment in routine practice provides additional information for oncologists to identify patients at risk of poorer prognosis and holds the potential to offer more personalized treatment planning including chemotherapy regimen adjustment, frequent monitoring to detect early signs of recurrence, and addressing inflammation or nutritional problems, to prolong survival.

This is the first study in Indonesia to investigate the prognostic significance of inflammatory and nutritional biomarkers, particularly CAR, in patients with stage I–IV breast cancer receiving chemotherapy. Previous studies recruited selective breast cancer patients (Zhou et al., 2019; Liu, Guo & Zhang, 2021; Kayikciouglu & Onder, 2022) while our study included patients with stage I–IV to cover various patients’ characteristics and yielded findings with broader applicability. This study also analyzed three survival outcomes including DFS, PFS, and OS, representing breast cancer recurrence, disease progression, and mortality. We compared predictive value among several well-known and comparably novel inflammatory and nutritional biomarkers, including NLR, PLR, LMR, SII, SIRI, PIV, PNI, and CAR, which were measured after surgery and before chemotherapy. Zhou et al. (2018) investigated the predictive values of inflammatory biomarkers before and after tumor resection in colorectal cancer patients. The post-surgery inflammatory biomarkers and their dynamic changes, particularly neutrophil and monocyte-to-lymphocyte ratio (NMLR), SII, and CAR were prognostic predictors of colorectal cancer. Inflammation from the surgical wound healing process ceased one-month post-surgical procedure. Thus, it was expected that there would be no surgery-related disruptions expressed in inflammatory biomarkers. Consequently, it is essential and practical to assess inflammatory biomarkers post-surgery.

There are several limitations worth noting in the present study. First, this study was a single-centre study with a limited sample size. Findings from a single Indonesian population may not apply broadly, thus a multicentric study would enhance applicability. Second, we generated and used specific cut-off values of inflammation and nutritional biomarkers which are study-specific. Its universal applicability across populations and cancer types warrants further validation study. These conditions demanded cautious interpretation and implementation for patients with different clinicopathological characteristics. Hence, further prospective studies are warranted to validate our findings.

Conclusions

CAR was demonstrated as a promising prognostic marker in stage I–IV breast cancer patients receiving chemotherapy. Among other investigated inflammatory and nutritional biomarkers, CAR showed the greatest discriminating ability. Based on our results, a high CAR (CAR > 1.5) at diagnosis may serve as a tool to identify patients at risk of unfavorable DFS and OS. This approach can guide personalized treatment planning including chemotherapy regimen adjustment, frequent monitoring to detect early signs of recurrence, and addressing inflammation or nutritional problems.

Supplemental Information

Supplemental Information 1 Raw data

The clinical, pathological, biomarkers, survival, and follow up duration data. The raw data consists of both numeric and categorical value.

Supplemental Information 2 Kaplan-Meier curves for 3-year disease-free survival based on inflammatory and nutritional biomarkers

(A) Neutrophil-to-lymphocyte ratio (NLR). (B) Platelet-to-lymphocyte ratio (PLR). (C) Lymphocyte-to-monocyte ratio (LMR). (D) Systemic immune-inflammation index (SII). (E) Systemic inflammation response index (SIRI). (F) Pan-immune inflammation value (PIV). (G) Prognostic nutritional index (PNI).

Supplemental Information 3 Kaplan-Meier curves for 3-year progression-free survival based on inflammatory and nutritional biomarkers

(A) Neutrophil-to-lymphocyte ratio (NLR). (B) Platelet-to-lymphocyte ratio (PLR). (C) Lymphocyte-to-monocyte ratio (LMR). (D) Systemic immune-inflammation index (SII). (E) Systemic inflammation response index (SIRI). (F) Pan-immune inflammation value (PIV). (G) Prognostic nutritional index (PNI).

Supplemental Information 4 Kaplan-Meier curves for 3-year overall survival based on inflammatory and nutritional biomarkers

(A) Neutrophil-to-lymphocyte ratio (NLR). (B) Platelet-to-lymphocyte ratio (PLR). (C) Lymphocyte-to-monocyte ratio (LMR). (D) Systemic immune-inflammation index (SII). (E) Systemic inflammation response index (SIRI). (F) Pan-immune inflammation value (PIV). (G) Prognostic nutritional index (PNI).

We gratefully thank Riani Witaningrum, Benedreky Leo, Irfan Haris, Norma Dewi Suryani, Betrix Rifana Kusuma, Refdiana Dewi, F Linda Tri Pramatasari, Efri Kurniawan, and Sumartiningsih for technical assistance.

Additional Information and Declarations

Competing Interests

Author Contributions

Human Ethics

Data Availability

The authors declare there are no competing interests.

Susanna Hilda Hutajulu conceived and designed the experiments, performed the experiments, analyzed the data, authored or reviewed drafts of the article, and approved the final draft.

Yufi Kartika Astari analyzed the data, prepared figures and/or tables, authored or reviewed drafts of the article, and approved the final draft.

Meita Ucche analyzed the data, authored or reviewed drafts of the article, and approved the final draft.

Nyoman Kertia conceived and designed the experiments, authored or reviewed drafts of the article, and approved the final draft.

Yanri Wijayanti Subronto conceived and designed the experiments, authored or reviewed drafts of the article, and approved the final draft.

Dewi Kartikawati Paramita analyzed the data, authored or reviewed drafts of the article, and approved the final draft.

Lina Choridah performed the experiments, authored or reviewed drafts of the article, and approved the final draft.

Ericko Ekaputra performed the experiments, authored or reviewed drafts of the article, and approved the final draft.

Irianiwati Widodo performed the experiments, authored or reviewed drafts of the article, and approved the final draft.

Suwardjo Suwardjo performed the experiments, authored or reviewed drafts of the article, and approved the final draft.

Mardiah Suci Hardianti conceived and designed the experiments, authored or reviewed drafts of the article, and approved the final draft.

Kartika Widayati Taroeno-Hariadi conceived and designed the experiments, authored or reviewed drafts of the article, and approved the final draft.

Ibnu Purwanto conceived and designed the experiments, authored or reviewed drafts of the article, and approved the final draft.

Johan Kurnianda conceived and designed the experiments, authored or reviewed drafts of the article, and approved the final draft.

The following information was supplied relating to ethical approvals (i.e., approving body and any reference numbers):

Medical and Health Research and Ethics Committee, Faculty of Medicine, Public Health and Nursing, Universitas Gadjah Mada (reference number: KE/FK/0417/EC/2018).

The following information was supplied regarding data availability:

The raw data is available in the Supplementary File.

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
