# Peer review of "Prognostic significance of C-reactive protein (CRP) and albumin-based biomarker in patients with breast cancer receiving chemotherapy"

_PeerJ, doi:10.7717/peerj.19319_

## Round 0.1 · original submission · Major Revisions

Dear Authors

Kindly resubmit by making all the changes recommended by the referees.

Reviewer 1 ·

Basic reporting

1. Authors are required to incorporate the most recent epidemiological data related to breast cancer in Indonesia and worldwide to ensure the accuracy, relevance, and contextual significance of their findings.
2. Authors must correct all grammar and syntax errors throughout the manuscript to ensure clarity, readability, and professional quality.
3.Authors should clearly emphasize how their study contributes new insights or advances beyond existing research in the related field. Highlighting the novel aspects, unique methodologies, or broader implications of the findings will demonstrate the study’s added value and relevance.
4. As the authors note in the discussion, the sample size of the study is small. To strengthen and validate their findings, they are encouraged to incorporate in silico analyses using publicly available datasets. Utilizing data from established repositories can enhance the robustness of the conclusions, provide external validation, and improve the overall impact of the study.

Experimental design

1. Certain details currently presented in the Results section would be more appropriately placed in the Methods section to enhance the manuscript’s structure and readability. For instance, baseline patient characteristics, biomarker cut-off values, and similar procedural information should be described in the Methods to avoid redundancy and clearly distinguish between the study design and its findings.
2. The manuscript should specify which patients received adjuvant, neoadjuvant, or palliative treatments, based on factors such as breast cancer stage, grade, age, etc. The rationale for these treatment decisions should also be clearly outlined in the Methods section. This information will provide context for the treatment strategies, enhance the transparency of the study design, and ensure the interpretability of the findings.
3. The Methods section should clearly indicate whether the chemotherapeutic regimens are adjuvant, neoadjuvant, or palliative, and include the specific names of the therapeutic drugs used. This will provide essential context for the treatment strategies and ensure a comprehensive understanding of the study design.
4. The Liu method used for determining the biomarker cut-off should be appropriately cited in the manuscript. Including the relevant citation will provide proper attribution and allow readers to reference the original methodology for a more comprehensive understanding of the approach.
5. The criteria for defining low vs. high C-reactive protein (CRP)-to-albumin ratio (CAR) should be clearly outlined, along with the rationale for these definitions. This will ensure transparency in the categorization of CAR levels and provide context for the interpretation of the study’s findings.
6. In lines 186–188, the authors note that high CAR was more frequent in patients with stage 4 cancer receiving neoadjuvant chemotherapy. The implications of this finding should be discussed in more detail. What might this suggest about the relationship between CAR levels and disease stage or treatment type? Does it imply that patients receiving adjuvant chemotherapy have lower CAR values, and if so, what are the clinical or biological implications of this difference? The comparison between these groups requires further elaboration to fully understand its significance.
7.It is unclear whether patients receiving neoadjuvant chemotherapy were further categorized based on pathological complete response (pCR) versus residual disease. Clarifying this distinction and examining any differences in CAR levels between these subgroups could provide valuable insights into the relationship between CAR levels and treatment response.
5. It is not clear from the Methods whether the analysis pertains to the primary tumor or metastatic lesions. If the study focuses solely on the primary tumor, the rationale for excluding metastatic sites should be clearly defined.

Validity of the findings

1. Certain sections in the Discussion, such as the presentation of study findings, should be moved to the Results section. This will help maintain a clear distinction between the presentation of data and its interpretation, improving the overall structure and flow of the manuscript.
2. In lines 236–240, the authors should provide an explanation for why this is the case. It is important to include the authors’ opinions, rationale, and the implications of these findings to give readers a clearer understanding of their significance. Colorectal cancer may not be the most appropriate comparison for this study. It would be more relevant to explore studies focused on breast cancer for a more accurate comparison. Are there existing studies that address similar comparisons in the context of breast cancer?
3. In lines 243–244, the authors mention adjuvant and palliative therapies but fail to include neoadjuvant therapy. This omission should be addressed to provide a more complete overview of the treatment strategies under discussion.
4. The message in lines 263–264 is unclear, particularly regarding the chemotherapy regimen used. Was there a variation in the chemotherapy treatments among stage 4 patients? If so, the rationale for these differences should be explained, as this information is currently missing.
5. Lines 273–285 should be moved to the beginning of the Discussion to emphasize the importance of CRP in breast cancer and provide the rationale for this study. This will help to establish a clearer context for the study’s objectives and significance.
6. The Discussion is currently difficult to follow and would benefit from restructuring. It should be reorganized to improve clarity and flow, ensuring that key points are presented logically. Consider breaking the discussion into clear sections that address the study’s findings, their implications, comparison with existing literature, and limitations, followed by suggestions for future research.
7. The figures need to be reformatted, as the font size is too small and makes them difficult to follow. Increasing the font size and ensuring clarity of labels and legends will improve the readability and overall quality of the figures.
8. It is crucial that each figure includes the sample size (n) for every category to provide proper context. The Y-axis labels must be clearly defined, as their current meaning is unclear. Additionally, p-values should be explicitly denoted in every figure to highlight statistical significance. The figure legends should be comprehensive and detailed, offering clear explanations of the data, methods, and any relevant annotations to enhance understanding and ensure clarity for the readers.

Additional comments

Major revisions are needed for the manuscript. Several sections require clarification, including the figures, discussion, and methodology. The figures should include sample sizes (n) for each category, clearer Y-axis labels, p-values, and more detailed legends. The discussion needs to be restructured for better flow and clarity, and the rationale for key aspects of the study should be more thoroughly explained. Overall, substantial improvements are necessary to enhance the manuscript’s clarity, structure, and comprehensiveness.

Reviewer 2 ·

Basic reporting

General comments:
• A high CRP-to-albumin ratio (CAR >1.5) is significantly linked to poorer disease-free survival (DFS) and overall survival (OS) in breast cancer patients undergoing chemotherapy, complementing traditional prognostic factors like age, stage, and HER2 status.

• As the first study in Indonesia to assess CAR and other biomarkers (e.g., NLR, PLR, SII) in breast cancer patients (stages I-IV), it addresses a gap in regional data and offers insights into local disease management.

Experimental design

General comments:
• Findings align with international studies on CAR’s impact but reveal lower DFS rates and hazard ratios in this population, suggesting potential demographic or treatment differences.

• CAR can help identify high-risk patients and guide personalized treatment, especially in resource-limited settings.

Validity of the findings

Authors have to address the following comments:

• Findings from a single Indonesian population may not apply broadly; a multicentric study would enhance applicability. In this context, it needs to be explained in the main text.

• CAR showed no correlation with progression-free survival (PFS), requiring further investigation into this discrepancy. In this regard, it needs to be elaborated in the main manuscript.

Additional comments

Authors have to address the following comments:

• The CAR threshold (1.5) differs from other studies (e.g., 3.85 in pancreatic cancer), raising questions about its universal applicability across populations and cancer types. In this context, it needs to be explained in the main text.

• The biological link between elevated CAR and poor prognosis remains unexplored, warranting studies on chronic inflammation and tumor aggressiveness. In this regard, it needs to be explained in the main text.

---

## Round 0.2 · accepted · Accept

Dear Authors
Congratulations
The referees have approved your revised version

Reviewer 1 ·

Basic reporting

The authors have done a commendable job in revising the manuscript. The quality of the figures can still be improved, if feasible. Additionally, if it aligns with the journal’s formatting guidelines, I suggest positioning the figure legends beside the figures rather than on the opposite side for better readability

Experimental design

The revised manuscript addresses all previous concerns

Validity of the findings

no comment

Additional comments

no comment. Manuscript is well revised.

Reviewer 2 ·

Basic reporting

The authors have addressed all the suggested comments. Before publication, they need to submit a complete tracked changes file highlighted in yellow and a clean file.

After submitting the highlighted file, this paper can be published in PeerJ journal.

Experimental design

Appropriately developed.

Validity of the findings

Has been presented well in the manuscript.

Additional comments

The authors have addressed all the suggested comments. Before publication, they need to submit a complete tracked changes file highlighted in yellow and a clean file.

After submitting the highlighted file, this paper can be published in PeerJ journal.